# Large-scale seroepidemiology uncovers nephro-urological pathologies in people with tau autoimmunity

Andreia D. Magalhães[¤a], Marc Emmenegger[¤b], Elena De Cecco, Manfredi Carta[¤c], Karl Frontzek[¤d], Andra Chincisan[¤e], Jingjing Guo, Simone Hornemann*, Adriano Aguzzi*

Institute of Neuropathology, University of Zurich, Zurich, Switzerland

¤a Current address: Department of Neurology, Inselspital, Bern University Hospital, Bern, Switzerland
¤b Current address: Division of Medical Immunology, Department of Laboratory Medicine, University Hospital Basel, Basel, Switzerland
¤c Current address: Institute of Neurology, University of Zurich, Zurich, Switzerland
¤d Current address: University College London, Institute of Neurology, Queen Square Brain Bank, London, United Kingdom
¤e Current address: Credit Suisse, Zurich, Switzerland
* simone.hornemann@uzh.ch (SH); adriano.aguzzi@uzh.ch (AA)

## Abstract

Intraneuronal aggregates of the microtubule-associated protein tau play a pivotal role in Alzheimer's disease and several other neurodegenerative syndromes. Anti-tau antibodies can reduce pathology in mouse models of neurodegeneration and are currently being tested in humans. Here, we performed a large-scale seroepidemiological search for anti-tau IgG autoantibodies ($\alpha\tau$) on 40,497 human plasma samples. High-titer $\alpha\tau^+$ individuals were surprisingly prevalent, with hospital patients being three times more likely to be $\alpha\tau^+$ ($EC_{50} \geq 2^6$; a nominal dilution of >1/64) than healthy blood donors (4.8% versus 1.6%). The prevalence increased with age over 70 years-old (RR 1.26, 95% CI 1.11–1.43, $P < 0.001$) and was higher for women (RR 1.20, 95% CI 1.07–1.39, $P = 0.002$). The autoantibodies bound selectively to tau, inhibited tau aggregation in vitro, and interfered with tau detection in plasma samples. No association was found between $\alpha\tau$ autoantibodies and neurological disorders. Instead, tau autoreactivity showed a significant association with kidney and urinary disorders (adjusted RR 1.27, 95% CI 1.10–1.45, $P = 0.001$ and 1.40, 95% CI 1.20–1.63, $P < 0.001$, respectively). These results suggest a previously unrecognized association between $\alpha\tau$ autoimmunity and extraneural diseases.

## Introduction

Tau is a microtubule binding protein involved in cytoskeletal dynamics and expressed in neurons [1,2] and in extraneural tissues, including kindey [1–5]. It plays a pivotal role in a variety of neurodegenerative diseases, including Alzheimer's disease (AD)

**Data availability statement:** All relevant data are provided in the main text or Supporting information. Raw blot image files are available in S1 Raw Images, and unprocessed raw images for Fig 2E are deposited on Figshare (https://doi.org/10.6084/m9.figshare.30405028). Raw patient data can be made available to qualified researchers upon request, in accordance with the conditions established by the Data Governing Board of the University Hospital Zurich (data-governance@usz.ch).

**Funding:** University of Zurich Foundation (UZH Candoc grant, FK-19-025 (https://www.research.uzh.ch/en/funding.html) to A.D.M.; Swiss Personalized Health Network (SPHN, driver project, 2017DRI17, https://sphn.ch) to A.A.; Schweizerischer Nationalfonds zur Förderung der Wissenschaftlichen Forschung (SNF, 179040, 207872, 183563, https://www.snf.ch) to A.A.; European Research Council (ERC, Prion2020, 670958ERC, https://www.euresearch.ch) to A.A.; NOMIS Stiftung (NOMIS Foundation, https://nomisfoundation.ch) to A.A.; Innovation Fund of the University Hospital Zurich (INOV00096, https://www.usz.ch/en/health-innovation-hub) to A.A.; University Hospital Zurich (Foundation grant, USZF27101, https://usz-foundation.com) to M.E., A.A.; HMZ ImmunoTarget grant (https://www.hochschulmedizin.uzh.ch) to A.A.; Stiftung Neuropath to A.A.; Fondazione Gelu (Gelu Foundation): A.A.; Michael J. Fox Foundation for Parkinson's Research (MJFF, MJFF-020710, MJFF-021073, https://www.michaeljfox.org) to S.H. The funders had no role in study design, data collection and analysis, decision to publish, or preparation of the manuscript.

**Competing interests:** The authors have declared that no competing interests exist.

**Abbreviations:** AD, Alzheimer's disease; aRR, adjusted risk ratios; CI, confidence interval; DAPI, 4,6-diamidino-2-phenylindole; DTT, dithiothreitol; EDTA, ethylenediaminetetraacetic acid; ELISA, Enzyme-Linked Immunosorbent Assay; HEPES,4-(2-hydroxyethyl)piperazine-1-ethanesulfonic acid; HRP, horseradish peroxidase; IQR, interquartile range; LPS, lipopolysaccharides; NMDAR, N-methyl-D-aspartate receptor; PBS, phosphate-buffered saline; PBST, phosphate-buffered saline 0.1% Tween20; PrP$^c$, cellular prion protein; RT, room temperature; ThT, Thioflavin T; TMB, 3,3′,5,5′-Tetramethylbenzidine.

[6], progressive supranuclear palsy, and various syndromes collectively referred to as tauopathies [7]. The presence of brain neurofibrillary tau tangles correlates with cognitive decline [8], and plasma measurements of total and phosphorylated tau have emerged as promising biomarkers for the detection and monitoring of AD progression [9–12]. Moreover, active and passive immunization with antibodies against a wide range of tau epitopes can reduce pathology and functional decline in animal models of tauopathies [13,14] and are being tested in clinical trials of neurodegenerative diseases [15].

Natural autoantibodies are immunoglobulins generated against self-antigens in the absence of external antigen stimulation [16]. They are a normal part of the immunoglobulin repertoire and have physiological roles in homeostasis and surveillance, including the clearance of cellular debris, anti-inflammatory activity, and first-line defense against pathogens [17]. However, in certain situations, natural autoantibodies can also cause pathological autoimmunity. The distinction between homeostatic and pathological autoantibodies is sometimes unclear and may depend on an individual's physiological state [18]. The study of natural autoantibodies can therefore inform about properties of their targets, e.g., unrecognized protective or contributing roles in disease [19]. Some natural autoantibodies can cause neurological disorders, such as antibodies targeting the N-methyl-D-aspartate receptor (NMDAR) in encephalitis [20] or antibodies targeting aquaporin-4 in neuromyelitis optica [21]. Conversely, natural antibodies against amyloid-β have been suggested to protect and slow the progression of AD; aducanumab, a monoclonal antibody developed from B-cells of cognitively normal older age individuals, has been studied as a treatment for AD [22].

Anti-tau autoantibodies have been detected in plasma of patients with AD [23] and Parkinson's disease [24], but also in non-neurodegeneration controls [23]. The effects, if any, of natural anti-tau autoantibodies in modulating the risk of developing neurodegenerative diseases are unknown. The study of individuals with anti-tau autoantibodies could clarify their potential as modifiers or biomarkers of disease. Here, we tested 40,497 plasma samples from healthy blood donors and from patients admitted to a university hospital in the frame of a two-sites cross-sectional study. We found that anti-tau autoimmunity is highly specific and surprisingly frequent, with its prevalence increasing with age. Unexpectedly, natural anti-tau autoantibodies were associated with a previously unrecognized syndrome comprising kidney and urinary disorders.

## Results

### Prevalence of naturally occurring plasma anti-tau autoantibodies

We designed a cross-sectional study to investigate naturally occurring plasma IgG autoantibodies against the microtubule binding domain of tau protein (MTBD-tau) corresponding to the human truncated 4R-tau (residues 244–372 relative to 2N4R human tau). Plasma samples from 32,291 university hospital patients (age ≥18 years) and 8,206 healthy blood donors were screened by miniaturized Enzyme-Linked Immunosorbent Assay (ELISA, Fig 1A) [25–27].

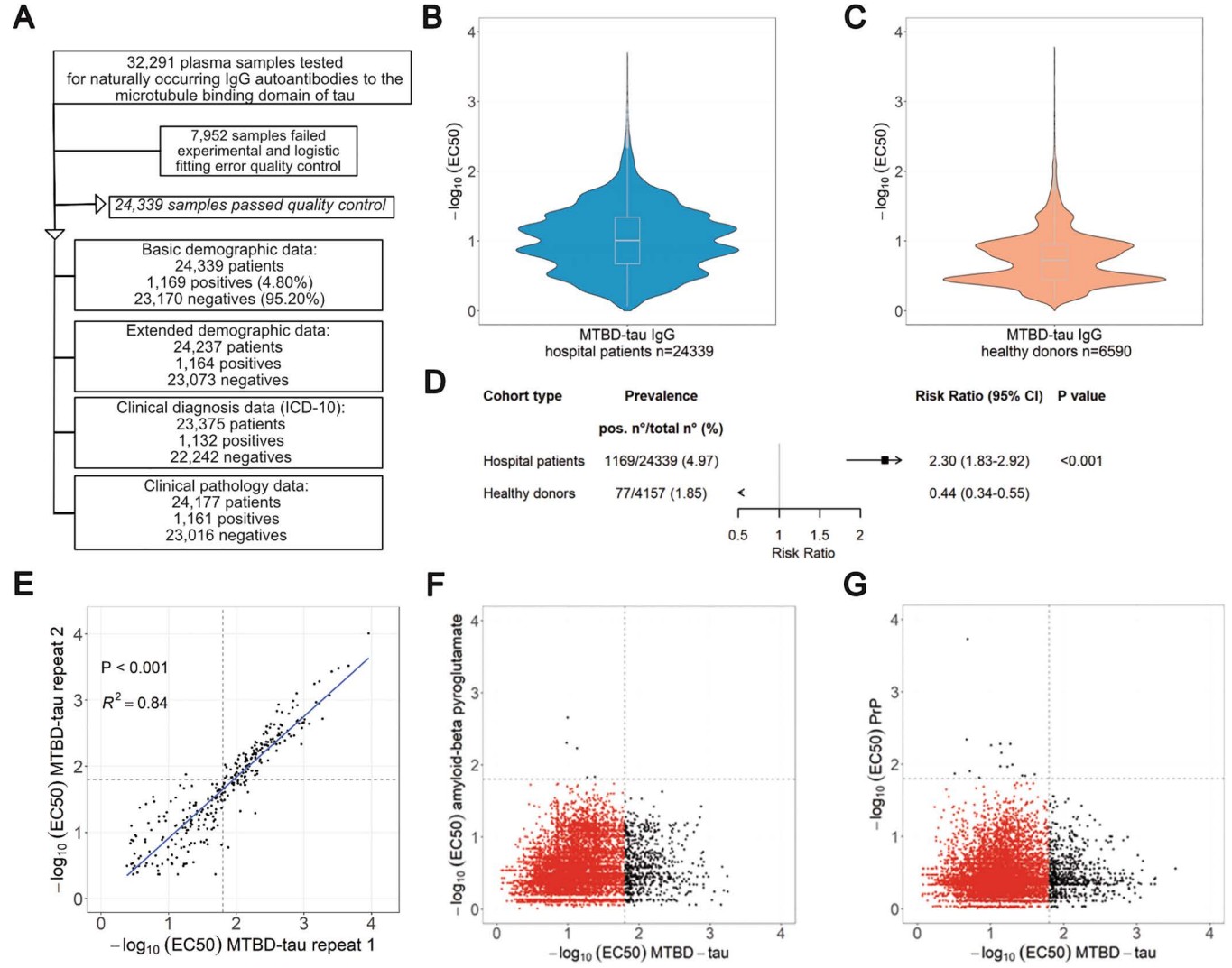

**Fig 1. Study overview and seroprevalence of anti-MTBD-tau IgG autoantibodies. (A)** Flowchart of the samples of the hospital patients' cohort. **(B, C)** Distribution of $-\log_{10}(EC_{50})$ values obtained from the microELISA screen of hospital (B) and blood donor (C) plasma samples. **(D)** Age- and sex-adjusted risk ratios and 95% confidence intervals (CI; I bars) for the detection of anti-tau autoantibodies in hospital and blood-bank plasma samples. **(E)** Replicability of microELISA duplicates with independent estimation of the $-\log_{10}(EC_{50})$ values. Dashed lines: cut-off value of $-\log_{10}(EC_{50}) = 1.8$. **(F)** $-\log_{10}(EC_{50})$ values of samples tested by microELISA against MTBD-tau and amyloid-β pyroglutamate. **(G)** Same as shown in (F), but for samples tested against MTBD-tau and the cellular prion protein (PrPᶜ). The underlying numerical data for panels (B), (C), (E), (F) and (G) can be found in S1 Data.

We excluded 7,952 and 1,616 non-informative samples from patients and blood donors, respectively (fitting error >20% $-\log_{10}(EC_{50})$ or high background), and analyzed 24,339 patient samples and 6,590 healthy blood-donor samples (Fig 1A–C). A titer of $-\log_{10}(EC_{50}) \geq 1.8$, approximately corresponding to a nominal dilution of >1/64 or an $EC_{50} \geq 2^6$, was empirically selected as a cutoff to call tau-autoreactive samples (henceforth named ατ⁺) [25]. This threshold represents the inflection point in our serial dilution assays above background and below saturation and was chosen to best balance sensitivity and specificity. 1,169 hospital samples (4.8%) but only 104 healthy donor samples (1.6%) were ατ⁺ (Fig 1A–C). Hence, anti-tau immunoreactivity was more prevalent in unselected hospital patients than in healthy

PLOS Biology

individuals ($P < 0.001$). Demographic data was available for 4,157 of the 6,590 blood donors. Median ages were 42 (IQR 29–54) and 55 years (IQR: 40–69) for healthy donors and hospital patients, respectively ($P < 0.001$). Of the healthy blood donors, 40.9% ($n = 1,698$) were women, whereas for the hospital group, 47.7% ($n = 11,609$) of the patients were women ($P < 0.001$). Multi-variate log-binomial regression [28,29] adjusted for age and sex showed that hospital patients had a 2.3× higher risk than healthy donors to be $\alpha\tau^+$ (adjusted risk ratio [aRR] 2.30, 95% confidence interval [CI] 1.83–2.92, $P < 0.001$, Fig 1D).

The replicability of the microELISA screen was found to be high ($R^2 = 0.84$, $P < 0.001$, Fig 1E). There was no cross-reactivity to two other proteins implicated in neurodegeneration, amyloid-β pyroglutamate and the cellular prion protein ($PrP^C$) [27]. Of 12,297 patient samples, 604 samples were positive against MTBD-tau and 5 against amyloid-β pyrogluta-mate but none was cross-reactive against both targets ($P = 1$, $\chi^2$ test) (Fig 1F), Moreover, of 13,099 patient samples, 694 were reactive against MTBD-tau and 15 against $PrP^C$, but again none was cross-reactive against both targets ($P = 0.734$, $\chi^2$ test) (Fig 1G).

## Specificity and biological activity of $\alpha\tau^+$ samples

To investigate specificity, we purified anti-tau autoantibodies by MTBD-tau affinity chromatography from four individual $\alpha\tau^+$ patients (P1–P4) and from a pool of six $\alpha\tau^+$ sample. Purified anti-MTBD-tau autoantibody samples had $EC_{50}$ val-ues of 0.288–14.45 µg/ml whereas the mouse anti-MTBD-tau antibody RD4 [30] had $EC_{50} = 0.002$ µg/ml (Fig 2A). In a soluble-competition immunoassay (Fig 2B), purified $\alpha\tau^+$ autoantibodies showed a concentration-dependent binding to recombinant MTBD-tau purified by cation exchange and size exclusion chromatography and to a pool of 8 synthetic MTBD-tau peptides but not to albumin or an unrelated synthetic peptide (Fig 2C). To probe for polyreactivity, purified $\alpha\tau^+$ autoantibodies were tested against several structurally unrelated autoantigens and bacterial proteins, including MTBD-tau, albumin, cardiolipin, DNA, insulin, and lipopolysaccharides (LPS). Anti-tau autoantibodies were reactive against MTBD-tau but not against any of the other antigens or uncoated plates (Fig 2D).

In immunofluorescent stainings, purified $\alpha\tau$ autoantibodies co-localized to cytoplasmic EGFP-0N4RTau in SH-SY5Y cells and showed similar binding patterns to an anti-tau mouse monoclonal antibody H7, but not to non-transfected cells (Fig 2E). On western blots, purified $\alpha\tau$ autoantibodies detected tau-specific bands in cell lysates of SH-SY5Y cells over-expressing tau$^{P301L/S320F}$, but not in parental (wt) SH-SY5Y cells (Fig 2F and S1 Raw Images). Hence, the immunoreactivity of $\alpha\tau^+$ samples was highly specific for tau. All IgG subclasses and both κ and λ light chains were present in $\alpha\tau^+$ patients' samples (Fig 2G and 2H). Epitope mapping and light-chain typing revealed a polytypic response in at least 8 out of the 13 $\alpha\tau^+$ samples (Fig 2H and 2I).

To investigate whether purified $\alpha\tau^+$ autoantibodies interfere with the aggregation of MTBD-tau, samples from patients P7 and P8 were tested in an in vitro tau aggregation assay [31]. MTBD-tau aggregation was induced by heparin and shak-ing and monitored using Thioflavin T (ThT; Fig 3A). The presence of $\alpha\tau^+$ autoantibodies reduced the plateau of the ThT fluorescence signal in the kinetic trace by about half, whereas antibodies purified by protein G affinity chromatography from an $\alpha\tau^-$ patient had no effect (Fig 3A). Hence, $\alpha\tau^+$ autoantibodies were able to specifically bind to and inhibit MTBD-tau aggregation in vitro.

Plasma tau is a promising biomarker of progression for several neurological diseases [9–12,32]. We examined the effects of $\alpha\tau$ autoantibodies onto the performance of a plasma tau immunoassay (Fig 3B). Purified $\alpha\tau$ autoantibodies from 5 patients (P2–P6) were added to tau$^{441}$-spiked plasma. After incubation, the amount of free tau$^{441}$ was analyzed by ELISA using commercial anti-tau antibodies for capture (BT2, epitope on human tau$^{441}$: residues 194–198) and detec-tion (ab64193, epitope on non-phosphorylated and phosphorylated human tau$^{441}$ surrounding residue 262) [33]. Purified $\alpha\tau$ autoantibodies P4–P6 induced a concentration-dependent impairment of detection of tau$^{441}$ hampering the detection of plasma-spiked tau$^{441}$ by approximately 10-fold at higher anti-MTBD-tau autoantibody concentration (Fig 3C and 3D). In contrast, P2–P3 did not show any significant impairment of the detection of plasma-spiked tau$^{441}$. This variability of interference is explained by the binding epitopes on tau$^{441}$. P4-P6 occupy tau residues mapping to the binding epitope of

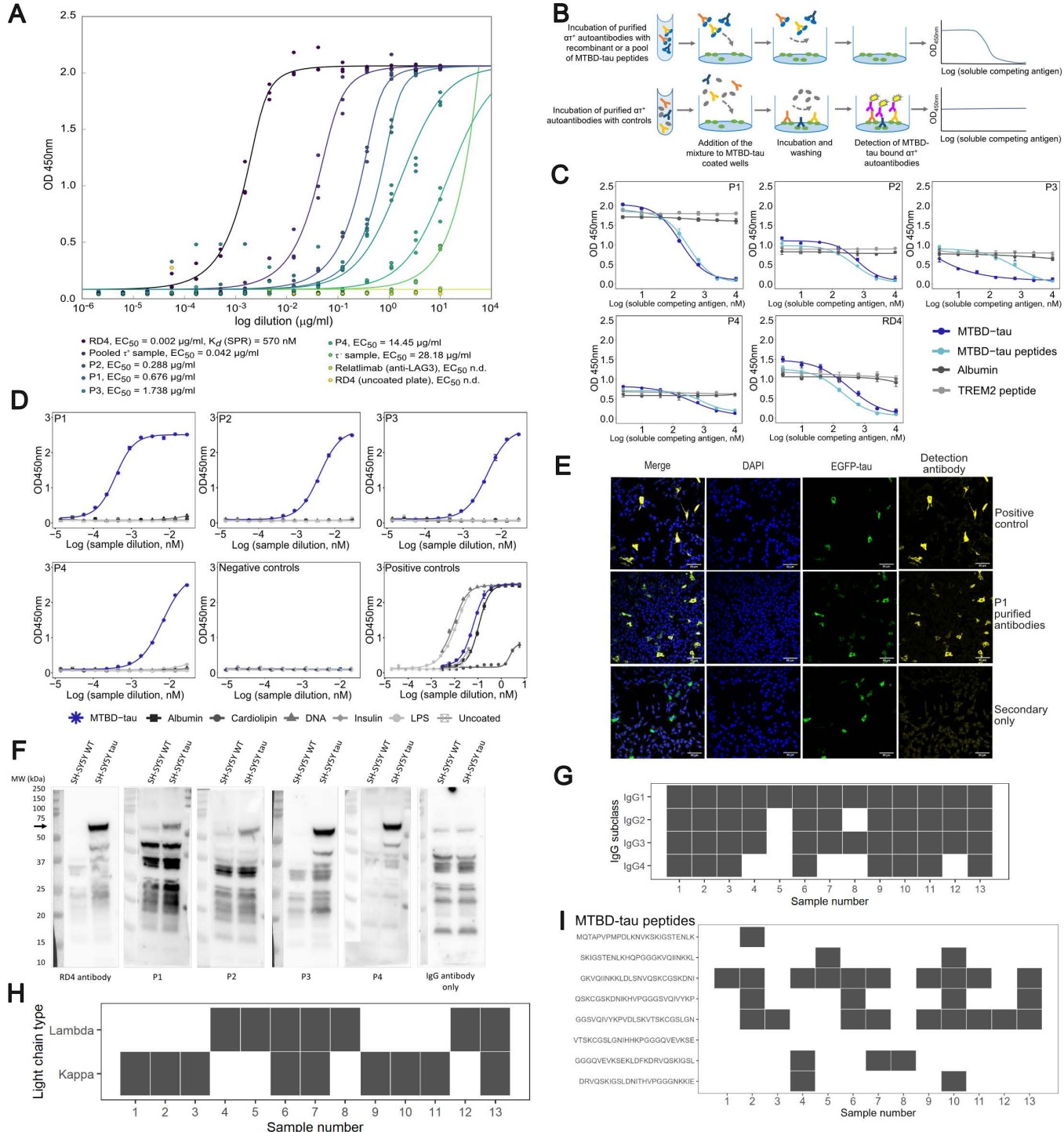

**Fig 2. Biophysical characterization of samples from αt⁺ patients. (A)** Indirect ELISA of purified anti-MTBD-tau autoantibodies from four ατ⁺ samples and from 6 pooled ατ⁺ samples. All antibodies were assayed at the same concentration to compare their $EC_{50}$ to that of RD4. Relatlimab (anti-LAG3) was used as negative control. $EC_{50}$ values are indicated in the figure. n.d.: not determined. **(B)** Principle of competition ELISA. **(C)** Competition ELISA of purified anti-tau autoantibodies from four ατ⁺ patients against albumin, recombinant MTBD-tau, a pool of synthetic peptides spanning the sequence of MTBD-tau and synthetic TREM2. RD4 antibody: positive control. Mean values ± SD of two replicates. **(D)** Indirect ELISA to assess the reactivity of purified anti-tau autoantibodies against albumin, cardiolipin, double-stranded DNA, insulin, lipopolysaccharides (LPS), MTBD-tau and uncoated plates.

Positive controls were as follows: RD4 antibody for MTBD-tau, anti-DNP antibody for cardiolipin, albumin and DNA and the IKC pool of 20 heparin plasma samples for insulin, LPS and uncoated plates. Mean values±SD of two replicates are shown. **(E)** Representative immunofluorescence images of SH-SY5Y cells expressing EGFP-0N4R-tau using affinity-purified anti-tau autoantibodies. HT7 pan-tau antibody: positive control. Secondary only as negative control: omission of primary antibody. Scale bar: 60 µm. **(F)** Western blot of cell lysates from wild-type (SH-SY5Y WT) or tau[P301L/S320F]-overexpressing cells (SH-SY5Y tau) using purified anti-tau autoantibodies from four ατ$^+$ patients. Positive control: RD4. IgG antibody only as negative control: omission of primary antibody. **(G)** IgG subclass typing of 13 ατ$^+$ plasma samples. Gray and white boxes: reactive and non-reactive samples, respectively. **(H)** κ/λ light chain typing of the samples in G. **(I)** Epitope mapping of the samples in G against 25mer MTBD-tau peptides with 10 residues overlap. Vertical axis: sequences of the MTBD-tau peptides covering the sequence of MTBD-tau. The underlying numerical data for panels (A), (C), (D), (G), (H) and (I) can be found in S1 Data. Raw images of the Western blots in (F) can be found in S1 Raw Images.

ab64193, whereas P2-P3 occupy residues mapping outside the epitope of ab64193 (Figs 3C and 3D and S1). Hence, the presence of ατ$^+$ autoantibodies can interfere with the detection of plasma tau in immunoassays depending on the combination of epitopes of the patient samples and of commercial antibodies used in immunoassays.

## Demographic characteristics of tau-immunoreactive patients

The age of ατ$^+$ patients (median: 58 years; IQR: 43−71) was significantly higher than that of ατ$^-$ (median: 55; IQR: 40−68; $P<0.001$, Fig 4A). The prevalence of ατ immunoreactivity increased with age, from 3.9% in patients aged <29 years to 7.6% in patients aged >90 years ($P<0.001$, Fig 4B). A log-binomial regression model [28,29] estimated that the RR for the presence of anti-MTBD-tau autoantibodies was highest for patients aged 70−99 years (RR 1.26, 95% CI 1.11–1.43, $P<0.001$, Fig 4C) and for women (RR 1.20, 95% CI 1.07–1.39, $P=0.002$, Fig 4C). Due to the association of anti-MTBD-tau autoantibodies with increased age and female sex, all further RRs were calculated using a multivariate log-binomial regression model adjusted for age and sex. To identify potential correlations between ατ$^+$ and specific diseases, we analyzed the admission to clinical departments. The highest RRs for ατ$^+$ were found for angiology (aRR 1.84, 95% CI 1.21–2.63, $P=0.002$) and nephrology (aRR 1.50, 95% CI 1.16–1.89, $P=0.001$, Fig 4D and 4E), whereas no significant difference was found between the percentage of ατ$^+$ and ατ$^-$ patients from the department of neurology (Fig 4D).

## Neurological disorders and anti-tau autoimmunity

We next mined pseudonymized clinical diagnoses from the clinical records of the USZ pertaining to the diagnoses of 23,375 patients as represented by ICD-10 codes (Fig 5A). Given the involvement of tau in neurodegenerative diseases [6], we focused on evaluating the association between anti-MTBD-tau autoantibodies and neurological diseases, which we categorized in 23 main groups of disorders. No associations between ατ$^+$ and neurological diseases were identified (Fig 5A). We further performed a targeted screen using plasma samples from 47 patients with AD and 68 similarly aged non-AD patients selected from our plasma biobank (median age of AD patients 78 years, IQR 70,5–86, and median age of control patients 81 years, IQR 71–85, S1 Table). Samples were tested for anti-MTBD-tau IgG autoantibodies, as in the primary screen, and additionally for anti-MTBD-tau IgA autoantibodies and anti-full-length-tau (tau441) IgG and IgA autoantibodies. No significant difference in reactivity was observed between the plasma samples from AD patients and non-AD controls in this small convenience cohort (Fig 5B), which is in line with the previous finding that the presence of autoantibodies targeting MTBD-tau is unrelated to AD or other neurological disorders.

## Systemic disorders and anti-tau autoimmunity

We next assessed possible associations between tau autoimmunity and extraneural diseases, which we binned into 27 main groups of disorders (Fig 6A). After adjustment for multiple comparisons, ατ immunoreactivity showed significant associations with vascular disorders (aRR 1.51, 95% CI 1.28–1.77, $P<0.001$), nutritional disorders (aRR 1.31, 95% CI 1.14–1.50, $P<0.001$), anemia (aRR 1.49, 95% CI 1.21–1.82, $P<0.001$), kidney disorders (aRR 1.27, 95% CI 1.10–1.45, $P=0.001$) and urinary disorders (aRR 1.40, 95% CI 1.20–1.63, $P<0.001$) (Fig 6A), whereas no association was observed

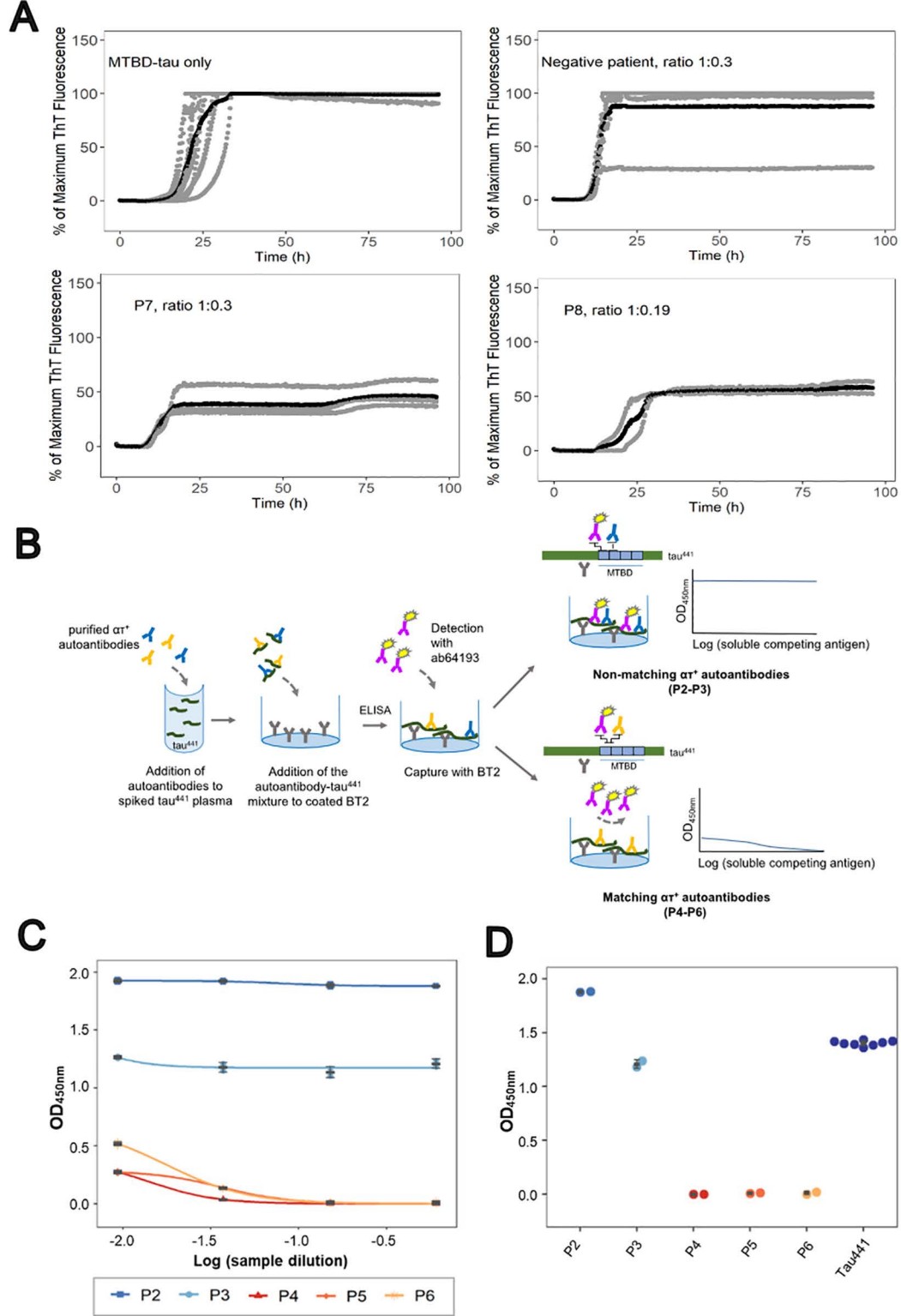

**Fig 3. Natural ατ autoantibodies inhibit tau aggregation and impair tau detection. (A)** Kinetic aggregation curves of MTBD-tau followed by ThT fluorescence in the absence ($n = 12$) or presence of purified ατ autoantibodies (patient P7 ($n = 4$) and P8 ($n = 2$)) or antibodies from an ατ-sample ($n = 8$). Stoichiometric ratios as indicated. Gray lines indicate individual replicates and black lines the average of the replicates. **(B)** Principle of the competition

sandwich ELISA. **(C)** Competitive sandwich ELISA to assess the ability of purified anti-tau autoantibodies with matching (P4–P6) and non-matching (P2 and P3) epitopes to the commercial detection antibody, ab64193, to impair the detection of free tau441 spiked in plasma (dark blue). Serial dilutions of purified anti-tau antibodies are shown in (C) and binding at highest anti-tau antibody concentrations in **(D)**. The underlying numerical values of panels (A), (C), and (D) can be found in S1 Data.

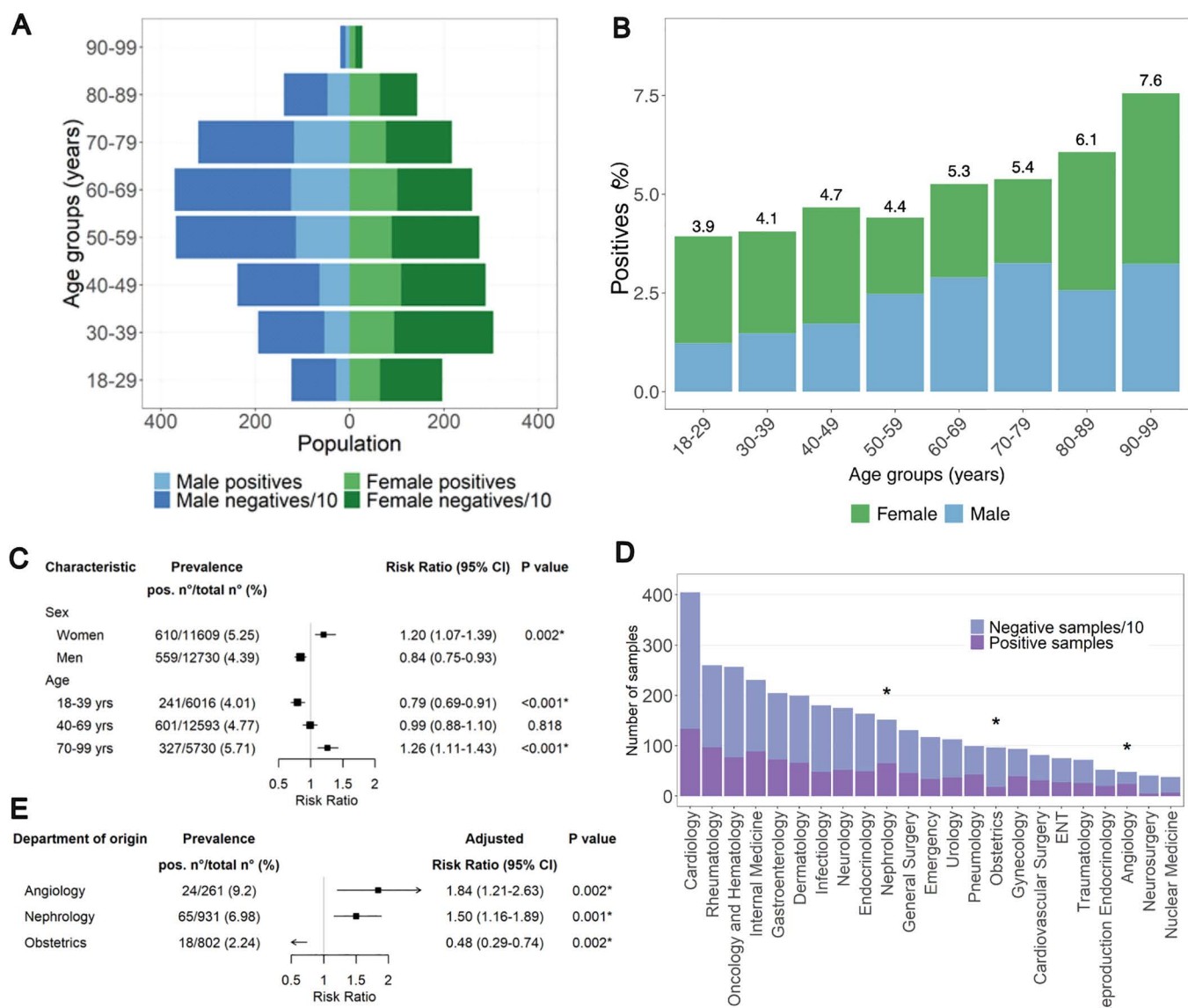

**Fig 4. Demographic characteristics of hospital patients' cohort. (A)** Age and sex pyramid of positive and negative individuals. **(B)** Percentage of positives among hospital patients across decadic age groups. The numbers on top of each bar correspond to the positivity rates. $\chi^2$ test for trend in proportions. **(C)** RR ± 95% CI for $\alpha\tau^+$ autoantibodies according to sex and age groups. Asterisks: $P < 0.05$ after Bonferroni correction. **(D)** Breakdown of samples by clinical department. Asterisks: $P < 0.05$ (two-proportions $z$-test with Bonferroni correction). **(E)** aRR ± 95% CI for $\alpha\tau^+$ samples by clinical department. Asterisks: $P < 0.05$ after Bonferroni correction.

## A

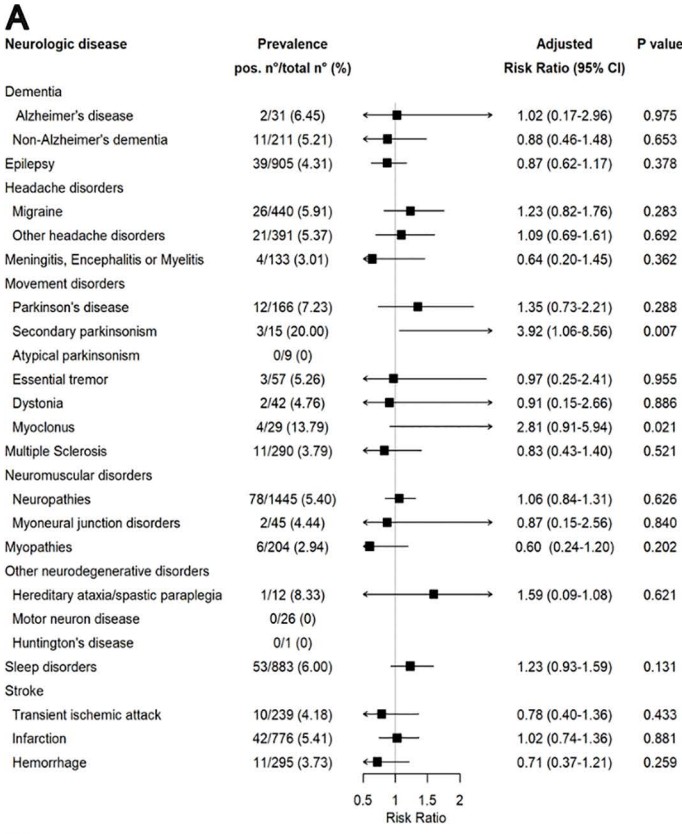

| Neurologic disease | Prevalence pos. n°/total n° (%) | | Adjusted Risk Ratio (95% CI) | P value |
|---|---|---|---|---|
| **Dementia** | | | | |
| Alzheimer's disease | 2/31 (6.45) | | 1.02 (0.17-2.96) | 0.975 |
| Non-Alzheimer's dementia | 11/211 (5.21) | | 0.88 (0.46-1.48) | 0.653 |
| **Epilepsy** | 39/905 (4.31) | | 0.87 (0.62-1.17) | 0.378 |
| **Headache disorders** | | | | |
| Migraine | 26/440 (5.91) | | 1.23 (0.82-1.76) | 0.283 |
| Other headache disorders | 21/391 (5.37) | | 1.09 (0.69-1.61) | 0.692 |
| **Meningitis, Encephalitis or Myelitis** | 4/133 (3.01) | | 0.64 (0.20-1.45) | 0.362 |
| **Movement disorders** | | | | |
| Parkinson's disease | 12/166 (7.23) | | 1.35 (0.73-2.21) | 0.288 |
| Secondary parkinsonism | 3/15 (20.00) | | 3.92 (1.06-8.56) | 0.007 |
| Atypical parkinsonism | 0/9 (0) | | | |
| Essential tremor | 3/57 (5.26) | | 0.97 (0.25-2.41) | 0.955 |
| Dystonia | 2/42 (4.76) | | 0.91 (0.15-2.66) | 0.886 |
| Myoclonus | 4/29 (13.79) | | 2.81 (0.91-5.94) | 0.021 |
| **Multiple Sclerosis** | 11/290 (3.79) | | 0.83 (0.43-1.40) | 0.521 |
| **Neuromuscular disorders** | | | | |
| Neuropathies | 78/1445 (5.40) | | 1.06 (0.84-1.31) | 0.626 |
| Myoneural junction disorders | 2/45 (4.44) | | 0.87 (0.15-2.56) | 0.840 |
| Myopathies | 6/204 (2.94) | | 0.60 (0.24-1.20) | 0.202 |
| **Other neurodegenerative disorders** | | | | |
| Hereditary ataxia/spastic paraplegia | 1/12 (8.33) | | 1.59 (0.09-1.08) | 0.621 |
| Motor neuron disease | 0/26 (0) | | | |
| Huntington's disease | 0/1 (0) | | | |
| **Sleep disorders** | 53/883 (6.00) | | 1.23 (0.93-1.59) | 0.131 |
| **Stroke** | | | | |
| Transient ischemic attack | 10/239 (4.18) | | 0.78 (0.40-1.36) | 0.433 |
| Infarction | 42/776 (5.41) | | 1.02 (0.74-1.36) | 0.881 |
| Hemorrhage | 11/295 (3.73) | | 0.71 (0.37-1.21) | 0.259 |

## B

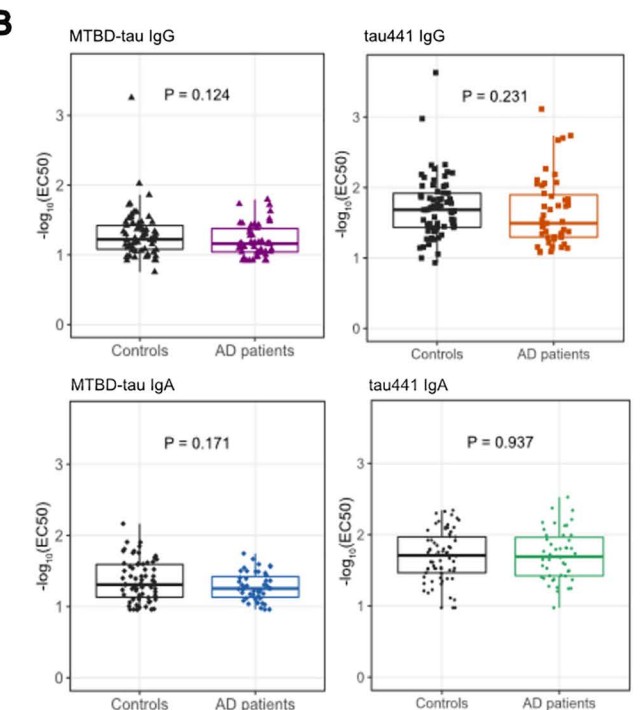

MTBD-tau IgG — P = 0.124 — Controls, AD patients

tau441 IgG — P = 0.231 — Controls, AD patients

MTBD-tau IgA — P = 0.171 — Controls, AD patients

tau441 IgA — P = 0.937 — Controls, AD patients

**Fig 5. aRR of MTBD-tau-autoreactivity for major groups of neurological disorders and reactivity profiles for AD screen. (A)** aRR and 95% CI (I bars) for $\alpha\tau^+$ autoantibodies according to different groups of neurological disorders. No *P* values remained significant after Bonferroni correction. **(B)** Box-plots showing the 25th, 50th (median), and 75th percentiles of the reactivity profiles for AD and control patients (Mann–Whitney *U* test). The underlying numerical values of panel (B) can be found in S1 Data.

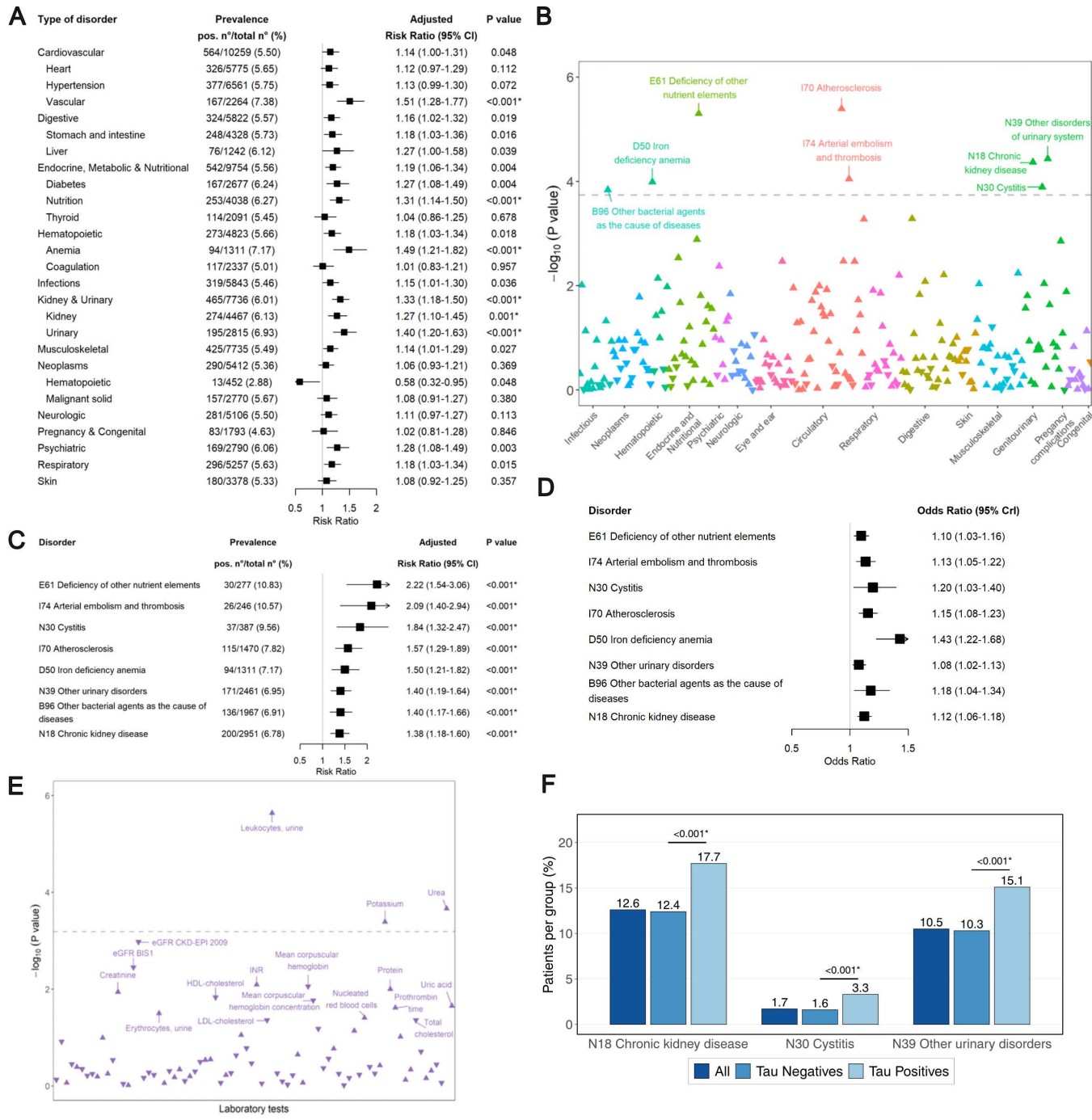

**Fig 6. Association of MTBD-tau IgG autoantibodies with systemic disorders and clinical laboratory data. (A)** aRR ± 95% CI (bars) for ατ⁺ auto-antibodies grouped 27 different main systemic clinical conditions. Asterisks: *P* < 0.05 after Bonferroni correction. **(B)** −log₁₀(*P* value) of the log-binomial regression for the presence of plasma MTBD-tau IgG autoantibodies for 276 disorders according to ICD-10 codes and adjusted for age and sex. Triangles pointing upwards and downwards: positive and negative coefficients, respectively; gray dashed line: *P* < 0.05 after Bonferroni correction (here and henceforth). *P* values significant after Bonferroni adjustment are labeled with the ICD codes. **(C)** Same as (A), but according to individual disease entities from (B). **(D)** Bayesian logistic regression adjusted for age and sex on ICD-10 codes significantly associated in B. and using −logit₁₀(EC₅₀) as continuous outcome (odds ratio ± 95% credible intervals). This analysis confirms the positive association of these ICD-10 codes with ατ reactivity (from B). **(E)** −log₁₀(*P* value) of the log-binomial regression for the presence of plasma MTBD-tau IgG autoantibodies and clinical laboratory parameters. *P* < 0.05 are labeled with the clinical laboratory parameters. Gray dashed line: *P* < 0.05 after Bonferroni correction. eGFR CKD-EPI 2009: estimated

glomerular filtration rate using the Chronic Kidney Disease Epidemiology Collaboration 2009 equation; eGFR BIS1: estimated glomerular filtration rate using the older adults Berlin Initiative Study 1 equation. INR: international normalized ratio. LDL: low-density lipoprotein. HDL: high-density lipoprotein. **(F)** Calculated prevalence of different ICD-10 codes in the total cohort (dark blue), ατ⁻ (intermediate blue) and ατ⁺ samples (light blue). Asterisks: $P < 0.05$ after Bonferroni correction (two-proportions z-test).

among patients with autoimmune disorders (S2 Fig). There was no difference in the coincidence of all comorbidities between ατ⁺ samples (5, 0.46%) and ατ⁻ samples (69, 0.33%) ($P = 0.633$).

To address potential biases arising from grouping and selecting major categories, we explored the association between ατ⁺ and individual ICD-10 codes. Across 276 individual ICD-10 codes and after correction for multiple comparisons, eight exhibited significant associations with ατ⁺ (Fig 6B and 6C). These included "E61–deficiency of other nutrient elements" (aRR 2.22, 95% CI 1.54–3.06, $P < 0.001$), "I74–arterial embolism and thrombosis" (aRR 2.09, 95% CI 1.40–2.94, $P < 0.001$), "N30–cystitis" (aRR 1.84, 95% CI 1.32–2.47, $P < 0.001$), "I70–atherosclerosis" (aRR 1.57, 95% CI 1.29–1.89, $P < 0.001$), "D50–iron deficiency anemia" (aRR 1.50, 95% CI 1.21–1.82, $P < 0.001$), "N39–other urinary disorders" (aRR 1.40, 95% CI 1.19–1.64, $P < 0.001$), "B96–other bacterial agents as the cause of diseases" (aRR 1.40, 95% CI 1.17–1.66, $P < 0.001$) and "N18–chronic kidney disease" (aRR 1.38, 95% CI 1.18–1.60, $P < 0.001$; Fig 6C). Conversely, we found no significant negative association between tau autoreactivity and any ICD-10 codes that could suggest a decreased risk for a specific disease in ατ⁺ patients (Fig 6B). To verify relevant associations, we used Bayesian logistic regression using logit-transformed $EC_{50}$ values ($-logit[EC_{50}]$) as a continuous outcome. All the previously referred eight ICD-10 codes showed positive associations with $-logit[EC_{50}]$ (Fig 6D).

To validate these conditions as significant associations with plasma MTBD-tau IgG autoantibodies, we examined the association between ατ⁺ and 106 commonly requested laboratory parameters (S2 Table) using data available for 24,177 patients, independently of any ICD-10 classifiers. After correction for multiple comparisons, the laboratory markers "Leukocytes, urine", "Potassium," and "Urea" were positively associated with tau autoimmunity (Fig 6E). The association with increasing levels of urea and potassium suggests a link to kidney failure, whereas leukocytes in urine are a feature of urinary tract infections. These findings support the hypothesis that anti-tau autoimmunity correlates with such disorders. Accordingly, the prevalence of chronic kidney disease was 12.4% in ατ⁻ and 17.7% in ατ⁺ patients, whereas that of other urinary disorders was 10.3% in ατ⁻ and 15.1% in ατ⁺ patients (Fig 6F).

## Discussion

This study was designed to discover novel associations between autoreactivity to tau and human diseases. This approach of "reverse immunopathology" is hypothesis-free: it does not pose constraints on the type of diseases that might result from anti-tau immunity. While the approach is generally applicable to any autoantigen, its predictive power relies on the analysis of large patient collectives, which in turn requires the development of high-throughput methods at reasonable cost. We therefore miniaturized antibody detection to 3 µl/sample in 1,536-well microplates, allowing to run 40,000 assays/24 h with minimal hands-on time. We performed >300,000 immunoassays on >40,000 plasma samples for ατ autoreactivity. Each sample was measured at eight dilution points, enabling precise and unambiguous titer determinations [25–27].

Plasma ατ IgG were more prevalent in women and increased with age, reaching 7.6% in the cohort of 90–99-year-olds. The specificity of ατ⁺ plasma to MTBD-tau was high and was confirmed by multiple assays including epitope mapping. The age- and sex-adjusted ατ⁺ rate in hospital patients was 3-fold higher than in healthy blood donors and considerably exceeded the previously reported seroprevalences of autoantibodies targeting other intracellular neuronal targets (0.4% and 2%) [34], suggesting a potential correlation with disease states.

However, we did not find any association between ατ⁺ and any neurological conditions including neurodegenerative diseases using both ICD-10 classification as well as a convenience AD and non-AD cohort of plasma samples, corroborating with reports of similar levels of plasma tau autoantibodies in AD patients and non-demented controls [35]. Instead, we uncovered a robust, specific association between ατ⁺ positivity and kidney and urinary disorders. This was independently validated by associations with higher prevalence of positive samples from the department of Nephrology, grouped kidney and urinary diseases, individual kidney and urinary diagnosis codes and related clinical laboratory biomarkers, such as high serum urea and potassium and high urine leukocytes.

Seroepidemiological studies cannot establish causality: ατ seropositivity could be a cause or a consequence of the clinical syndromes associated with it [36]. Therefore, due to its design, this study cannot conclude on the homeostatic or pathological role of the anti-tau IgG autoantibodies identified in the plasma of patients and healthy blood donors. However, tau is expressed in extraneural tissues, including kidney podocytes and urinary bladder [2–5], and its ablation causes glomerular pathologies [37].

These findings raise the possibility that plasma ατ autoantibodies might drive kidney and/or urological pathologies. Alternatively, they could reflect a physiological autoimmunity with low-affinity autoantibodies playing a role in immune surveillance and the clearance of debris. Another possibility is that an underlying pathological process causing renal/bladder tissue injury and inflammation, may lead to the extracellular release of intracellular proteins, including tau. Damage to tubular epithelial cells or glomerular structures may expose intracellular tau to the immune system, potentially breaking tolerance and triggering the emergence of natural or low-affinity IgG autoantibodies against tau. Renal dysfunction may also impair clearance of circulating antibodies or immune complexes, allowing even normally subclinical reactivity to become detectable. If the association between tau autoantibodies and kidney/urinary disease is epiphenomenal rather than causal these autoantibodies may represent useful biomarkers of these diseases. Therefore, it will be important to monitor renal and urinary function in the current clinical trials of tau immunotherapy [38–40].

Recent studies have explored the use of tau measurements in plasma as biomarkers for AD diagnosis and for monitoring its progression [9–12,32,41–45]. However, we found that anti-MTBD-tau autoantibodies could hinder tau detection in plasma by binding to epitopes recognized by commercial biomarker immunoassays. The extent of this interference likely depends on both the epitopes present and the design of the diagnostic assay, varying with the specific tau analytes targeted in the assay and the particular epitopes recognized by the patients' autoantibodies. This aligns with prior research indicating that both plasma anti-tau autoantibodies and administered anti-tau antibodies can influence the dynamics of tau levels in plasma [46]. Additionally, a recent community cohort study using an immunoassay for AD screening revealed associations between plasma tau levels and numerous comorbidities, with chronic kidney disease showing one of the strongest associations [47]. These results imply that the presence of anti-tau autoantibodies in plasma might impact the effectiveness of plasma tau as an AD biomarker. This consideration may become crucial as plasma tau levels move toward routine clinical use in AD diagnosis.

## Limitations

Our large-scale assessment of plasma tau autoimmunity has certain limitations. Firstly, most samples analyzed in our study came from a university hospital cohort, implying a bias towards complex pathologies and polymorbidity. To account for this, we included a vast collection of samples from healthy blood donors. Secondly, our study is confined to two sites within a single region with approximately 1,500,000 inhabitants whose ethnic composition was not thoroughly examined. Additionally, a direct comparison of the prevalence of plasma anti-tau antibodies in these two cohorts may potentially suffer from variations in materials, collection methods (e.g., addition of heparin that may cause aggregation of MTBD-tau), and handling processes at the two different sites. Thirdly, our study is restricted to the analysis of specific disease groups, the absence of an independent replication cohort and the lack of a longitudinal disease design. These constraints restrict our ability to verify our findings, assess temporal dynamics and the origin of these autoantibodies, and explore

potential causal relationships, including reverse causality, such as the presence of tau antibodies in patient cohorts with renal diseases. Fourthly, the detected anti-tau antibodies may crossreact with proteins expressed in the kidney or urinary tract, which could contribute to the observed association with kidney and urinary disorders. To explore this possibility, we performed an *in silico* analysis using BLASTP (https://blast.ncbi.nlm.nih.gov/) and compared the MTBD-tau to proteins in the human kidney and urinary proteome (UniProt/Swiss-Prot). This analysis identified residues 368–461 of nicotin-amide phosphoribosyltransferase (NAMPT) as a region with notably high sequence identity to MTBD-tau (~28%; S3 Fig). NAMPT is a metabolic enzyme with broad tissue distribution but relatively low expression in the kidney and urinary tract [2]. Nevertheless, such sequence-based analysis is limited detecting linear or conformational epitopes that might mediate cross-reactivity. Furthermore, the statistical analysis of the association of MTBD-tau-autoreactivity and different disorders was based on ICD-10, a medical classification focused on reimbursement and cause-of-death statistics [48]. We therefore performed additional statistical analyses using laboratory parameters, which have provided further support for the association of tau autoimmunity and systemic disorders. Finally, we used bacterially expressed MTBD-tau as target, thereby excluding full-length, other isoforms (such as big tau, a high molecular weight (~110 kDa) tau isoform that includes exon 4a in the MAPT transcript [49]) or post-translationally modified tau epitopes, like the phosphorylated forms p-tau181 [50] and p-tau217 [51], that are increasingly used as biomarkers for AD diagnosis. As a result, our methodology may miss plasma anti-tau autoantibodies associated with AD pathology and could underestimate the prevalence of total anti-tau autoantibodies. Furthermore, this approach may introduce a bias towards detecting autoantibodies targeting total periph-eral tau which is more abundant in plasma [52]. Therefore, larger studies with well-powered AD patient cohorts may be needed to more robustly assess the prevalence of autoantibodies selectively targeting brain-derived tau and to clarify their potential association with neurodegeneration.

## Conclusions

Our study identified a high seroprevalence of anti-MTBD-tau IgG autoantibodies in both plasma samples from university hospital patients and healthy blood donors. Tau autoimmunity is associated with female sex, older age, and previously unrecognized extraneural diseases. These findings point to unrecognized roles for tau and anti-tau autoantibodies in extraneural pathologies.

## Materials and methods

### Methods

**Study approval.** Collection of samples and clinical data were conducted according to study protocols approved by the Cantonal Ethics Committee of the Canton of Zurich, Switzerland (KEK-ZH Nr. 2015-0561, BASEC-Nr. 2018-01042, and BASEC-Nr. 2020-01731), in accordance with the provisions of the Declaration of Helsinki and the Good Clinical Practice guidelines of the International Conference on Harmonisation. All human donors and patients included in this study provided a written general informed consent.

**Study design.** From December 2017 until February 2020, residual heparin plasma samples were obtained from the Department of Clinical Chemistry, University Hospital of Zurich, USZ, Switzerland. Samples were collected during routine clinical care from patients admitted either as inpatients or outpatients (age ≥ 18 years) and were only included if basic demographic data was available, and an informed consent for research had been provided. From March until July 2020, EDTA plasma samples from blood donors were obtained from the Blood Donation Center of Zurich, Switzerland, according to standard criteria of blood donation. Exclusion criteria were as reported [25]. Plasma samples from patients of both sexes were examined in this study. Plasma samples were biobanked locally and tested in an automated indirect microELISA ([25–27] and below) for natural IgG autoantibodies against the MTBD-tau. Demographic and clinical data for the hospital cohort were obtained from clinical records of the USZ with follow-up until December 2021, while detailed clinical data for the blood donor cohort were not available for this study. ICD-10 codes [53] were used for clinical data

assessment. AD patients were selected using ICD-10 code F00 or G30. Non-neurodegeneration controls were defined by the lack of any Fxx or Gxx ICD-10 codes.

**Automated microELISA screen.** Plasma samples were tested for natural anti-MTBD-tau IgG autoantibodies in a microELISA screen [25–27]. Briefly, high-binding 1,536-well microplates (Perkin Elmer, SpectraPlate 1536 HB) were coated with 1 µg/mL of recombinant MTBD-tau (37 °C, 60 min). Plates were washed 3× with phosphate-buffered saline 0.1% Tween20 (PBST) using a Biotek El406 washer-dispenser and blocked with 5% milk (Migros)-PBST for 90 min. Plasma samples were diluted 1:20 in 1% milk-PBST and dispensed into the MTBD-tau-coated plates using ultrasound dispensing with an ECHO 555 Liquid Handler (Labcyte/Beckman Coulter). Each sample was tested at eight serial 2-fold dilutions (1:50–1:6,000) using different volumes to a final volume of 3 µL/well. Human IgG-depleted serum (MyBioSource) was used as negative and anti-tau RD4 (4-repeat isoform) mouse monoclonal antibody [30] (05–804 clone 1E1/A6 Merck Millipore) as positive control. Plates were incubated for 120 min at room temperature (RT) and washed 5x with PBST. Secondary antibody peroxidase AffiniPure goat anti-mouse IgG H + L (115-035-003, Jackson ImmunoResearch) 1:2,000 diluted in 1% milk-PBST for the RD4 positive control, peroxidase AffiniPure goat anti-human IgG Fcγ-specific (109-035-098, Jackson ImmunoResearch) 1:4,000 diluted in 1% milk-PBST for the plasma samples and the IgG-depleted serum negative control were dispensed into the plates using a Biotek MultifloFX dispenser. Plates were incubated for 60 min, at RT and washed 3× with PBST. 3,3′,5,5′-Tetramethylbenzidine (TMB) Chromogen Solution for ELISA (Invitrogen) was added as colorimetric horseradish peroxidase (HRP) substrate for 3 min at RT. Finally, 0.5 M $H_2SO_4$ was added to stop the reaction. Plates were briefly centrifuged after each dispensing step except after dispensing of TMB. Plates were read at Optical Density = 450 nm ($OD_{450nm}$) in a plate reader (Perkin Elmer, Envision).

For the replicability assessment, 308 samples were tested in duplicates, running the replicates on the same day but using different 1536-well assay (destination) plates, different plate coordinates for each replicate, and calculating the −$\log_{10}(EC_{50})$ of each replicate independently.

Non-specific cross-reactivity was assessed by testing plasma samples against MTBD-tau and amyloid-β pyroglutamate (12,297 hospital patients' plasma samples) and the $PrP^C$ (13,099 hospital patients' samples) using a similar protocol as described above.

**Production and purification of recombinant tau.** The gene encoding the human truncated 4R-tau corresponding to MTBD-tau was cloned into a pRSET-A plasmid (Invitrogen) and expressed in *Escherichia coli* BL21(DE3) cells. Cultures were grown in Luria Broth (LB, Invitrogen) at 37 °C, induced with 1 mM isopropyl-β-D-thiogalactoside (IPTG) at an $OD_{600}$ of 0.8 and grown for additional 6 h at 37 °C before harvesting by centrifugation (6,000*g*, 10 min, 4 °C). Pellets were suspended and sonicated (30 min, 4 °C) in 20 mM piperazine-N,N-bis(2-ethanesulfonic) acid (PIPES), pH 6.5, 1 mM ethylenediaminetetraacetic acid (EDTA) and 50 mM 2-mercaptoethanol. After addition of NaCl to a final concentration of 500 mM, samples were boiled (95 °C, 20 min) and centrifuged (9,000*g*, 30 min, 4 °C). Ammonium sulfate was slowly added to a final concentration of 55% m/v and the suspension was stirred (1 h, RT). Samples were centrifuged (15,000*g*, 10 min, 4 °C), pellets were resuspended in 20 mM 4-(2-hydroxyethyl)piperazine-1-ethanesulfonic acid (HEPES), pH 7.0, 2 mM dithiothreitol (DTT), passed through a 0.45 µm Acrodisc filter (Sigma), and loaded onto Sepharose SP Fast Flow resin (Cytiva). Tau was eluted using a linear salt gradient from 0 to 1 M NaCl in 20 mM HEPES, pH 7.0, 2 mM DTT. Fractions containing tau were concentrated using Amicon Ultra-15 centrifugal filter unit (10-kDa MWCO, Merck) and dialyzed overnight at 4 °C against phosphate-buffered saline (PBS, pH 7.4, Kantonsapotheke Zurich), 1 mM DTT. Pooled samples were passed through a HiLoad 26/60 Superdex75 (GE Healthcare) column. Protein samples were analyzed by SDS-PAGE and samples containing tau were concentrated using an Amicon Ultra-15 centrifugal filter unit (10-kDa MWCO). Samples were assessed by SDS-PAGE and electrospray ionization-mass spectrometry. Pure MTBD-tau samples were stored until further use at −80 °C. The concentration of tau was determined using a bicinchoninic acid assay (Pierce BCA Protein Assay Kit, Thermo Fisher).

For the purification of full-length tau, a similar protocol was used with the following changes. The gene encoding the longest 4R isoform of human full-length tau protein, tau[441], (tau/pET29b, Addgene #16316, gift from Peter Klein [54]) was cloned into a pRSET-A plasmid (Invitrogen). For protein expression, *E. coli* BL21(DE3)pLysS cells were transformed with the pRSET-A plasmid encoding tau[441]. Cells were grown in Overnight Express Instant TB media (Novagen) for 6 h at 37 °C and then for 12 h at 25 °C. Fractions containing full-length tau were concentrated using Amicon Ultra-15 centrifugal filter unit (30-kDa MWCO, Merck).

**Purification of anti-tau autoantibodies from patient samples.** Heparin plasma (3−20 mL) was diluted 1:3.3 in PBS, and centrifuged at 6,000*g* for 10 min at 4 °C. The supernatant was loaded onto 3 mL of epoxy-MTBD-tau (prepared by overnight incubation of MTBD-tau and epoxy resin in 0.1 M $NaH_2PO_4$–NaOH, 1 M NaCl, pH 9.2) by repetitive loading of the plasma sample overnight at 4 °C. After washing with 50 mL of PBS, MTBD-tau autoantibodies were eluted 4× with 5 mL 0.1 M glycine–HCl, pH 2.5, and immediately neutralized to pH 7.0 with 1 M Tris-HCl, pH 8.5. Anti-MTBD-tau antibody-containing fractions were identified by indirect ELISA and concentrated stepwise using Amicon Ultra-15, Ultra-4, and Ultra-0.5 mL centrifugal filter units (50-kDa MWCO) up to a volume of 1 mL.

**Competitive ELISA.** For the competitive sandwich ELISAs for the detection of tau, high-binding 384-well plates (Perkin Elmer, SpectraPlate 384 HB) were coated with 4 µg/mL BT2 tau monoclonal antibody (#MN1010, Thermo Fisher) in PBS. After coating, plates were washed 3× with PBST and then blocked with 5% SureBlock (Lubio) in PBS for 180 min at RT. Recombinant human tau[441] (rPeptide) was diluted to a final concentration of 0.015 ng/mL and incubated with purified anti-tau autoantibodies 4-fold serially diluted (1:1.66 to 1:106.6) in 1:2 plasma under rotation at 500 rpm for 120 min at 37°C. Samples were transferred to BT2-coated plates and incubated for 45 min at RT. After washing 4× with PBST, plates were incubated with ab64193 (polyclonal IgG antibody; Abcam, 0.125 µg/mL) for 45 min at RT. Plates were washed 4x with PBST and incubated with peroxidase AffiniPure goat anti-Rabbit IgG (H + L) (111-035-045, Jackson ImmunoResearch), at a 1:2,000 dilution for 60 min at RT. Plates were washed 4× with PBST and 1-Step Ultra TMB-ELISA solution (Thermo Fisher) was added for 7 min at RT. After addition of 0.5 M $H_2SO_4$, plates were read at $OD_{450}$ nm in a plate reader (Perkin Elmer, Envision).

For the competitive ELISAs of MTBD-tau autoantibodies, high-binding 384-well plates were coated with 20 µL of 0.5 µg/mL of MTBD-tau overnight at 4 °C. Afterward, plates were washed 3× with PBST and blocked with 5% SureBlock (Lubio) in PBST for 120 min. Purified autoantibodies from hospital cohort patients' plasma were diluted 1:50 in 1% SureBlock in PBST (sample buffer) and the anti-tau RD4 mouse monoclonal antibody to a final concentration of 0.4 µg/mL. Bovine serum albumin (BSA, Thermo Scientific), in-house purified recombinant MTBD-tau, a pool of eight synthetic peptides covering the sequence of MTBD-tau with 25 amino acids length and 10 amino acids of overlap (Genscript) and an unrelated 25 amino acid length synthetic TREM2 (Triggering receptor expressed on myeloid cells 2) peptide (GenScript) were used as competing antigens. Antibody samples were incubated overnight at 4 °C with serial 2-fold dilutions of antigen solutions in sample buffer, ranging from 20,000 to 2.44 nM. The antibody-antigen mixtures were then added to the plates and incubated for 45 min at RT. Plates were washed 3x with PBST, followed by the addition of secondary antibodies: peroxidase AffiniPure goat anti-Human IgG (H + L) (109-035-088, Jackson ImmunoResearch) at 1:3,000 dilution and peroxidase AffiniPure goat anti-mouse IgG (H + L) (115-035-003, Jackson ImmunoResearch) at 1:2,000 dilution. Secondary antibodies were incubated for 60 min at RT and plates were then washed 4× with PBST. TMB Chromogen Solution for ELISA (Invitrogen) was added to the plates and incubated for 7 min at RT. After addition of 0.5 M $H_2SO_4$, plates were read at $OD_{450nm}$ in a plate reader (SpectraMax Paradigm, Molecular Devices).

**Indirect ELISAs.** To test for polyreactivity, purified anti-tau autoantibodies were tested by indirect ELISA against several antigens [55,56]. High-binding 384-well plates were coated overnight at 4 °C with 20 µL of 1 µg/mL of MTBD-tau, 10 µg/mL of DNA from calf-thymus (Sigma), 10 µg/mL of LPS from *E. coli* O111:B4 (Sigma), 5 µg/mL of human insulin (Sigma), 10 µg/mL of BSA, 2 µg/mL of cardiolipin solution from bovine heart (Sigma), or left uncoated. Plates were washed 3× with PBST and then blocked in 5% SureBlock (Lubio) in PBST for 120 min at RT. Patient purified

anti-MTBD-tau autoantibodies were diluted 1:33, IgG-depleted plasma (BioSource) 1:50 diluted and used as negative control, anti-DNP (Sigma) diluted to 6 µg/mL, anti-tau RD4 to 6 µg/mL, and pooled plasma from 20 patients diluted 1:25 in 1% SureBlock in PBST as positive controls. Samples were serially diluted 12 times 1:1 with 1% SureBlock in PBST in the referred plates. Samples were incubated for 120 min at RT and washed 4× with PBST. 20 µL/well of secondary antibodies diluted in 1% SureBlock in PBST were added as follows: peroxidase AffiniPure goat anti-human IgG (H+L) (109-035-088, Jackson ImmunoResearch) diluted 1:3,000 and added to purified MTBD-tau autoantibodies and IKC pool wells, peroxidase AffiniPure goat anti-mouse IgG (H+L) (115-035-003, Jackson ImmunoResearch) at 1:2,500 dilution and added to anti-RD4 wells, and peroxidase AffiniPure goat anti-rabbit IgG (H+L) (111-035-045, Jackson ImmunoResearch) at 1:4,000 dilution and added to anti-DNP wells. Plates were then washed 4× with PBST. 20 µL of TMB Chromogen solution for ELISA (Invitrogen) was added and incubated for 7 min at RT. After addition of 0.5 M $H_2SO_4$, plates were read at $OD_{450nm}$ in a plate reader (SpectraMax Paradigm, Molecular Devices).

For relative affinity measurements, we used an indirect ELISA in 384-well plates (Perkin Elmer, SpectraPlate 384 HB) with MTBD-tau as a coating antigen using the following parameters: The starting concentration of all antibodies, assayed in triplicates, was 10 µg/mL. They were successively diluted 1:3 to reach a concentration of $2 \times 10^{-6}$ µg/mL. As reference antibody, we used purified RD4 kindly provided by Prof. Rohan de Silva (UCL Queen Square Institute of Neurology, London, UK). As additional controls, we used anti-LAG3 antibody Relatlimab [57], $\alpha\tau^-$ plasma sample, and uncoated plates. The respective $EC_{50}$ values were then determined using logistic regression, as above, and the curves as well as the dots were visualized.

For IgG subclassing, the following secondary antibodies were used: rabbit anti-human IgG1 (SA5-10202, Invitrogen), rabbit anti-human IgG2 (SA5-10203, Invitrogen), rabbit anti-human IgG3 (SA5-10204, Invitrogen) or rabbit anti-human IgG4 (SA5-10205, Invitrogen) at 1:1,500 dilution, peroxidase AffiniPure goat anti-rabbit IgG (H+L) antibody (111-035-045, Jackson ImmunoResearch) at 1:2,500 dilution.

For immunoglobulin light chain typing, the following secondary antibodies were used: Goat anti-Human Kappa-HRP [58] and Goat anti-Human Lambda-HRP [58] at 1:4,000 dilution and peroxidase AffiniPure goat anti-mouse IgG (H+L) (115-035-003, Jackson ImmunoResearch) at 1:2,500 dilution.

For the epitope mapping experiments, a similar approach was used in which high-binding 384-well plates were coated with 20 µL of 1 µg/mL of each MTBD-tau peptide (GenScript) in PBS overnight at 4°C. Plasma samples and human IgG-depleted serum (MyBioSource) used as negative control were diluted 1:50 and anti-tau RD4 mouse monoclonal antibody used as positive control (05−804 clone 1E1/A6 Merck Millipore) was diluted to a final concentration of 6 µg/mL in 1% SureBlock in PBST. The following secondary antibodies were used: peroxidase AffiniPure goat anti-human IgG (H+L) (109-035-088, Jackson ImmunoResearch) at 1:5,000 dilution and peroxidase AffiniPure goat anti-mouse IgG (H+L) (115-035-003, Jackson ImmunoResearch) at 1:2,500 dilution.

For data analysis of IgG subclassing, immunoglobulin light chain typing, and epitope mapping experiments, samples were considered reactive when the $OD_{450nm}$ was higher than the average of all negatives $OD_{450nm}$ +2× the standard deviation of the $OD_{450nm}$ of the negatives.

**Western blotting.** SH-SY5Y wild-type (Sigma) and cells overexpressing double-mutated tauP301L/S320F were lysed in 0.5% Triton X-100 (Sigma Aldrich) in PBS, supplemented with cOmplete Mini EDTA-free Protease Inhibitor Cocktail (Roche) on ice and supernatant was recovered after centrifugation at 14,000$g$ for 20 min at 4 °C. Protein concentration was determined using a bicinchoninic acid assay (Pierce BCA Protein Assay Kit, Thermo Fisher) and sample volumes were adapted to 30 µg of total protein. Samples were loaded onto NuPAGE 12% Bis-Tris gels (Invitrogen) and blotted to nitrocellulose membranes (Invitrogen) using a dry iBlot 2 Gel Transfer Device (Invitrogen, Thermo Fisher). Membranes were cut vertically along the protein ladder and blocked with 5% SureBlock in PBST at 30 min for RT and incubated overnight at 4 °C with patient-purified anti-MTBD-tau autoantibodies diluted 1:100 in 1% SureBlock, PBST. Anti-tau RD4 mouse monoclonal antibody diluted 1:4,000 was used as a positive control. Negative control membranes were not

incubated with primary antibodies. Membranes were washed 3× for 5 min with PBST and incubated with the following secondary antibodies for 60 min at RT: peroxidase AffiniPure goat anti-mouse IgG (H + L) (115-035-003, Jackson ImmunoResearch) and peroxidase AffiniPure goat anti-human IgG (H + L) (109-035-088, Jackson ImmunoResearch) 1:10,000 diluted. Membranes were washed 4× for 5 min with PBST and developed using the Immobilon Crescendo HRP Substrate (Millipore). Imaging was performed with the Fusion SOLO S imaging system (Vilber).

**Immunofluorescence.** SH-SY5Y cells were transfected with pRK5-EGFP-0N4Rtau plasmid (Addgene # 46904, a kind gift of Dr Karen Ashe [59]) using Lipofectamine 2000 according to manufacturer's protocol. After 48 h cells were fixed with 4% paraformaldehyde for 20 min at RT, permeabilized with 0.5% BSA, 0.1% Triton X-100 in PBS for 20 min at RT, and blocked with 0.5% BSA in PBS for 60 min at RT. For the immunocytochemical assays, 1:25 purified ani-MTBD-tau autoantibodies diluted in 0.5% BSA in PBS were incubated with the cells for 60 min. Tau mouse monoclonal antibody HT7 (MN1000 Thermo Fisher Scientific) diluted 1:500 was used as a positive control. Cells were washed 3× with 0.5% BSA in PBS and incubated for 30 min at RT with: goat anti-mouse IgG (H + L) cross-adsorbed Alexa Fluor 555 (A-21422 Invitrogen) diluted 1:500 and counterstained with 4,6-diamidino-2-phenylindole (DAPI) 1 mg/mL (Thermo Fisher) diluted 1:10,000 in 0.5% BSA in PBS, and for human antibody and negative control stains with biotin AffiniPure goat anti-human IgG (H + L) (109-065-003, Jackson ImmunoResearch) diluted 1:200 in 0.5% BSA in PBS. Samples were 3x washed with 0.5% BSA in PBS. For human antibody stains, samples were further incubated for 30 min at RT with streptavidin Alexa Fluor 594 conjugate diluted 1:200 and counterstained with DAPI diluted 1:10,000 in 0.5% BSA in PBS. Cells were mounted on glass slides with Fluorescence Mounting Medium (Thermo Fisher Scientific) and imaged using a Leica TCS SP5 confocal laser scanning microscope. Imaging was performed with equipment maintained by the Center for Microscopy and Image Analysis, University of Zurich.

**In vitro MTBD-tau aggregation assay.** MTBD-tau in vitro aggregation experiments were performed as previously described [31]. Briefly, 7 µM of in-house purified recombinant MTBD-tau, 3.5 µM of heparin (Santa Cruz Biotechnology) and 10 µM of ThT (Sigma) were diluted in PBS. The patient purified anti-tau autoantibodies and controls (plasma sample reactive against the LAG3 and IgG-purified using Protein G Sepharose (Cytiva)) were added at the indicated apparent stoichiometries in Fig 3. The mixtures with a total volume of 200 µl were added to black 96-well polystyrene microplates (Nunc, Prod. No. 265301) and ThT fluorescence (450/480 nm ex/em filters; bottom read mode) was measured at 37 °C under continuous orbital shaking (425 cpm) every 15 min for 96 h using a FLUOstar Omega microplate reader (BMG Labtech). The mean baseline fluorescence values were subtracted from the mean fluorescence values at each time point which were then normalized to maximum baseline-subtracted fluorescence values and multiplied by 100 [60].

**Statistical analysis.** Antibody titers were defined as the negative logarithm half-maximal responses ($-\log_{10}(EC_{50})$) obtained by fitting the $OD_{450nm}$ of the eight dilution points of each sample tested in the microELISA to a logistic regression fitter. We classified as positives samples with a cut-off of $-\log_{10}(EC_{50}) \geq 1.8$ [25], corresponding to a nominal dilution of >1/64. Non-informative samples (fitting error >20% $-\log_{10}(EC_{50})$ or high background) were excluded from the analysis [25]. In cases where more than one sample was available for the same individual the most recent $\log_{10}(EC_{50})$ value was used.

Age is presented as median with interquartile range (IQR) and comparisons were performed using non-parametric Mann–Whitney $U$ test. Categorized age groups and sex of positives and negatives are shown as percentages and compared using two-proportions $Z$-test or $\chi^2$ test for trend in proportions. Log-binomial regression models [28,61] using MTBD-tau autoreactivity were employed to estimate age- and sex-aRRs and 95% CIs, and to investigate the association between the detection of anti-MTBD-tau IgG autoantibodies and different demographic features.

For the AD cohort microELISA screen, $\log_{10}(EC_{50})$ values were compared using Mann–Whitney $U$ test. Additionally, Bayesian logistic regression was conducted [25,61–63], using $-\text{logit}_{10}(EC_{50})$ values as outcome, i.e., without dichotomizing the outcome using the R package rstanarm and the following priors (prior=normal[0, 2.5, autoscale=TRUE],prior_intercept = normal[5000, 2.5, autoscale=TRUE]) prior_aux = exponential [1, autoscale =TRUE].

We aimed to confirm the positive association between ICD-10 codes previously identified using a conventional logistic regression model with high $-\log_{10}(EC_{50})$ values. Each ICD-10 code was analyzed in an independent logistic regression and adjusted for age and sex.

The association between MTBD-tau-autoreactivity and neurological and systemic disorders, respectively, was analyzed by applying multivariate log-binomial regression models to estimate aRRs and 95% CIs. For neurological disorders, 23 major groups of ICD-10 codes (S3 Table) corresponding to neurological disorders identified at least once in the positive samples were used. For systemic disorders, the 27 major groups of ICD-10 codes (S4 Table) or 276 individual ICD-10 codes corresponding to systemic disorders identified at least once in the positive samples and with at least 200 total counts were used to avoid overinterpretation of rare cases of disease. Individual disease entities with a $P$ value <0.05 Bonferroni corrected for multiple comparisons were included in a multivariate log-binomial regression analysis.

For the association between MTBD-tau-autoreactivity and clinical laboratory parameters, we used laboratory parameters with more than 2,000 total counts and calculated median values of the total values available for each patient in case of repetition of the clinical laboratory test. We used multivariate log-binomial regression models to estimate aRR and 95% CI using 106 clinical laboratory tests.

Statistical significance was defined by two-tailed $P$-value ≤ 0.05. Statistical analyses and data visualization were performed using R version 4.3.2 and RStudio version 1.4.1106 [64].

## Supporting information

**S1 Table. Targeted AD screen samples.**
(DOCX)

**S2 Table. Laboratory parameters used in the statistical analysis.**
(DOCX)

**S3 Table. ICD-10 codes used for the grouping of neurological disorders in the statistical analysis.**
(DOCX)

**S4 Table. ICD-10 codes used for the grouping of systemic disorders in the statistical analysis.**
(DOCX)

**S1 Data. The numerical values of all replicates in Figs 1B**, **1C**, **1E**—**1G**, **2A**, **2C**, **2D**, **2G**—**2I**, **3A**, **3C**, **3D**, and **5B**.
(XLSX)

**S1 Raw Images. Raw images for the Western blots presented in Fig 2F.**
(PDF)

**S1 Fig. Epitope mapping of patient samples shown in Fig 3.** Epitope mapping of the same samples used in the assay shown in Fig 3C and 3D against eight 25mer MTBD-tau peptides overlapping 10 residues.
(TIFF)

**S2 Fig. RR for tau autoantibodies in autoimmune diseases.** Forest plot showing the risk ratios and 95% CI (I bars) for the detection of tau autoantibodies in plasma samples of patients according to ICD-10 diagnosis of 6 different autoimmune diseases. aRR and 95% CI were estimated using log-binomial regression multivariate models including the respective variable, age, and sex.
(TIFF)

**S3 Fig. Sequence alignment comparing the sequence of MTBD-tau (residues 13–114) to human NAMPT (residues 368–461; P43490).** The alignment results from a BLASTP search with the BLOSUM62 substitution matrix

(https://blast.ncbi.nlm.nih.gov/). The search was conducted against proteins present in the human kidney and urinary proteome (UniProt/Swiss-Prot). The alignment revealed 28% sequence identity.
(PDF)

## Acknowledgments

The authors wish to thank the hospital patients and blood donors for their generous altruistic contributions to this study. Imaging was performed with equipment maintained by the Center for Microscopy and Image Analysis, University of Zurich, Switzerland, and electrospray ionization-mass spectrometry at the Functional Genomics Center Zurich, University of Zurich, Switzerland. We thank Prof. Rohan de Silva (UCL Queen Square Institute of Neurology, London, UK) for providing the purified anti-MTBD-tau antibody RD4. We thank Dr. Marco Losa for the generous provision of secondary antibodies for light chain typing and Magdalena Bialkowska, Lisa Caflisch, Berre Doğançay, Julie Domange, Marigona Imeri, Lorène Mottier, Rea Müller, Antonella Rosati, Dezirae Schneider, and Anne Wiedmer for help with the high-throughput assays. Insightful advice about programming in R software was provided by Reto Guadagnini.

## Author contributions

**Conceptualization:** Andreia D. Magalhães, Simone Hornemann, Adriano Aguzzi.

**Data curation:** Andreia D. Magalhães, Marc Emmenegger, Manfredi Carta, Karl Frontzek, Andra Chincisan, Jingjing Guo, Simone Hornemann.

**Formal analysis:** Andreia D. Magalhães, Marc Emmenegger.

**Funding acquisition:** Andreia D. Magalhães, Marc Emmenegger, Simone Hornemann, Adriano Aguzzi.

**Methodology:** Andreia D. Magalhães, Marc Emmenegger, Elena De Cecco, Simone Hornemann, Adriano Aguzzi.

**Project administration:** Simone Hornemann, Adriano Aguzzi.

**Resources:** Marc Emmenegger, Elena De Cecco, Simone Hornemann, Adriano Aguzzi.

**Software:** Andreia D. Magalhães, Andra Chincisan.

**Supervision:** Simone Hornemann, Adriano Aguzzi.

**Validation:** Andreia D. Magalhães.

**Writing – original draft:** Andreia D. Magalhães.

**Writing – review & editing:** Andreia D. Magalhães, Marc Emmenegger, Elena De Cecco, Manfredi Carta, Karl Frontzek, Andra Chincisan, Jingjing Guo, Simone Hornemann, Adriano Aguzzi.

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
