## [Editor Report · Decision Letter 0]

1 Oct 2025

Dear Dr Aguzzi,

Thank you for submitting your revised manuscript from Review Commons entitled "Large-scale seroepidemiology uncovers nephro-urological pathologies in persons with tau autoimmunity" for consideration as a Research Article by PLOS Biology.

Your manuscript has now been evaluated by the PLOS Biology editorial staff, as well as by an academic editor with relevant expertise, and I am writing to let you know that the Academic Editor was able to arbitrate the revision and we do not think it requires a second look by the reviewers.

Once your full submission is complete, your paper will undergo a series of checks in preparation for the next step (minor pending some journal policies). To provide the metadata for your submission, please Login to Editorial Manager (https://www.editorialmanager.com/pbiology) within two working days, i.e. by Oct 03 2025 11:59PM.

Kind regards,

Melissa

Melissa Vazquez Hernandez, Ph.D.

Associate Editor

PLOS Biology

---

## [Editor Report · Decision Letter 1]

7 Oct 2025

Dear Dr Aguzzi,

Thank you for your patience while we considered your revised manuscript from Review Commons "Large-scale seroepidemiology uncovers nephro-urological pathologies in persons with tau autoimmunity" for publication as a Initial Research Submission at PLOS Biology. This revised version of your manuscript has been evaluated by the PLOS Biology editors, and an Academic Editor with relevant expertise.

Based on our Academic Editor's assessment of your revision, we are likely to accept this manuscript for publication, provided you satisfactorily address the remaining editorial requests. Please also make sure to address the following data and other policy-related requests.

1) We routinely suggest changes to titles to ensure maximum accessibility for a broad, non-specialist readership, and to ensure they reflect the contents of the paper. In this case, we would suggest a minor edit to the title, as follows. Please ensure you change both the manuscript file and the online submission system, as they need to match for final acceptance:

"Large-scale seroepidemiology uncovers nephro-urological pathologies in people with tau autoimmunity"

2) Please include information about the form of consent (written/oral) given for research involving human participants. All research involving human participants must have been approved by the authors' Institutional Review Board (IRB) or an equivalent committee, and must have been conducted according to the principles expressed in the Declaration of Helsinki.

3) The Ethics statement needs to be a separate, independent (and the first) subheading in the Material & Methods section. It must include the full name of the IACUC/ethics committee that reviewed and approved the animal care and use, as well as the protocol/permit/project license number. https://journals.plos.org/plosbiology/s/ethical-publishing-practice

Please supply the numerical values either in the a supplementary file or as a permanent DOI’d deposition for the following figures:

Figure 1BCEFG, 2ACD, 3ACD, 4A-E, 5AB, 6A-F, S3

5) Please cite the location of the data clearly in all relevant main and supplementary Figure legends, e.g. “The data underlying this Figure can be found in S1 Data” or “The data underlying this Figure can be found in https://doi.org/10.5281/zenodo.XXXXX”

6) Supplementary files (e.g., excel). Please ensure that all data files are uploaded as 'Supporting Information' and are invariably referred to (in the manuscript, figure legends, and the Description field when uploading your files) using the following format verbatim: S1 Data, S2 Data, etc. Multiple panels of a single or even several figures can be included as multiple sheets in one excel file that is saved using exactly the following convention: S1_Data.xlsx (using an underscore).

7) We will require these files before a manuscript can be accepted so please prepare and upload them now. Please carefully read our guidelines for how to prepare and upload this data: https://journals.plos.org/plosbiology/s/figures#loc-blot-and-gel-reporting-requirements

8) Deposition in a publicly available repository. Please also provide the accession code or a reviewer link so that we may view your data before publication. Please ensure that your Data Statement in the submission system accurately describes where your data can be found.

-- Please note that per journal policy, we do not allow the mention of "data not shown", "personal communication", "manuscript in preparation" or other references to data that is not publicly available or contained within this manuscript. Please either remove mention of these data or provide figures presenting the results and the data underlying the figure(s).

We expect to receive your revised manuscript within two weeks.

*Published Peer Review History*

*Press*

Sincerely,

Melissa

Melissa Vazquez Hernandez, Ph.D.

Associate Editor

PLOS Biology

---

## [Editor Report · Decision Letter 2]

27 Oct 2025

Dear Dr Aguzzi,

Thank you for the submission of your revised Research Article "Large-scale seroepidemiology uncovers nephro-urological pathologies in people with tau autoimmunity" for publication in PLOS Biology. On behalf of my colleagues and the Academic Editor, Mikael Simons, I am pleased to say that we can in principle accept your manuscript for publication, provided you address any remaining formatting and reporting issues. These will be detailed in an email you should receive within 2-3 business days from our colleagues in the journal operations team; no action is required from you until then. Please note that we will not be able to formally accept your manuscript and schedule it for publication until you have completed any requested changes.

PRESS

Sincerely, 

Melissa

Melissa Vazquez Hernandez, Ph.D., Ph.D.

Associate Editor

PLOS Biology
